# Targeting GRK5 for Treating Chronic Degenerative Diseases

**DOI:** 10.3390/ijms22041920

**Published:** 2021-02-15

**Authors:** Federica Marzano, Antonio Rapacciuolo, Nicola Ferrara, Giuseppe Rengo, Walter J. Koch, Alessandro Cannavo

**Affiliations:** 1Department of Advanced Biomedical Sciences, University of Naples Federico II, 80131 Naples, Italy; federica.marzano@outlook.com (F.M.); rapacciu@unina.it (A.R.); 2Department of Translational Medical Sciences, University of Naples Federico II, 80131 Naples, Italy; Nicola.ferrara@unina.it (N.F.); Giuseppe.rengo@unina.it (G.R.); 3Istituti Clinici Scientifici Maugeri—Scientific Institute of Telese Terme (BN), 82037 Telese Terme, Italy; 4Center for Translational Medicine, Temple University, Philadelphia, PA 19140, USA

**Keywords:** cardiovascular, neurodegeneration, cancer, GPCR, GRK5

## Abstract

G protein-coupled receptors (GPCRs) are the largest family of cell-surface receptors and they are responsible for the transduction of extracellular signals, regulating almost all aspects of mammalian physiology. These receptors are specifically regulated by a family of serine/threonine kinases, called GPCR kinases (GRKs). Given the biological role of GPCRs, it is not surprising that GRKs are also involved in several pathophysiological processes. Particular importance is emerging for GRK5, which is a multifunctional protein, expressed in different cell types, and it has been found located in single or multiple subcellular compartments. For instance, when anchored to the plasma membrane, GRK5 exerts its canonical function, regulating GPCRs. However, under certain conditions (e.g., pro-hypertrophic stimuli), GRK5 translocates to the nucleus of cells where it can interact with non-GPCR-related proteins as well as DNA itself to promote “non-canonical” signaling, including gene transcription. Importantly, due to these actions, several studies have demonstrated that GRK5 has a pivotal role in the pathogenesis of chronic-degenerative disorders. This is true in the cardiac cells, tumor cells, and neurons. For this reason, in this review article, we will inform the readers of the most recent evidence that supports the importance of targeting GRK5 to prevent the development or progression of cancer, cardiovascular, and neurological diseases.

## 1. Introduction

G protein-coupled receptor (GPCR) kinases (GRKs) are a family of serine/threonine (Ser/Thr) protein kinases that interact and phosphorylate agonist-activated GPCRs, thus triggering their desensitization and internalization within clathrin-coated vesicles [1,2,3,4]. Due to their important role, GRKs modulate almost all of the biological and pathophysiological aspects of GPCR signaling. To date, seven known members of the GRK family have been identified and divided into three different subfamilies, depending on their sequence homology and structural similarities: the rhodopsin kinases subfamily, which includes GRK1 and GRK7, the β-adrenergic receptor kinase (βARK) subfamily, comprising GRK2 and GRK3, and the GRK4-like subfamily, including GRK4, GRK5, and GRK6 [5,6,7]. Essentially, almost all the GRK isoforms are expressed ubiquitously with varying degrees of expression depending on the type of tissue or cell. For instance, GRK1 and GRK7 expression are restricted to the retina, where they primarily act as rhodopsin and cone opsin kinases [8,9]. GRK4 is highly expressed in the testes [10,11], brain [12], kidneys [13], and in the uterine myometrium [14]. Pertinent to this review is the evidence of a prominent GRK5 expression in the human heart and the limbic system (in the brain) [15]. Therefore, the role of this kinase has been widely investigated in both cardiovascular and neurological disorders. Moreover, since its expression is increased in tumor cells, several studies have suggested the inhibition of GRK5 as a novel potential therapeutic strategy against cancer. GRKs possess several protein domains, that affect their subcellular localization (e.g., plasma membrane, cytosol, mitochondria, and nucleus) [16]. For this reason, GRK5, and other GRKs [17], have been considered “multifunctional” and a double-edged sword protein able to interact with GPCRs and non-GPCR-related factors, depending on their location and specific stimuli. When at the plasma membrane, GRK5, like all other GRKs, exerts “canonical” effects, regulating the function of GPCRs. Accordingly, almost all of the seven GRK isoforms are localized in specific subcellular compartments that afford them closeness and access to GPCRs. While GRK2 and GRK3 are cytosolic proteins that translocate to the membrane in an agonist-dependent manner dependent on C-terminal (CT) lipid-binding moieties [6,18], GRK4, 5, and 6 are more basally located at the membrane. For decades, a lot of effort has been spent to define these classical actions. However, extensive work has revealed the existence of “non-canonical” and GPCR-independent roles for GRKs. For GRK2, these activities involve localization within mitochondria [6,19,20]. GRK2 appears to normally regulate mitochondria in a manner dependent on extracellular-signal-regulated kinase (ERK)-mediated phosphorylation and chaperone-binding. However, when GRK2 is accumulated into the mitochondria (e.g., after and in ischemic stress), this kinase participates in the derangement of metabolism, by altering fatty-acid-mediated oxygen consumption, and stimulates the generation of reactive oxygen species (ROS) generation [6,20]. GRK5 has non-canonical activity within the cytosol or nucleus, where it can interact with several non-GPCR-related proteins or with DNA, becoming a major ruler of cell-growth and apoptosis (please see sections below).

Despite the plethora of effects attributed to GRK5, the role played by this kinase in governing mammalian physiology and pathology is still under intensive investigation. Accordingly, it is very important to study the so-called “GRK interactome”, which, as shown below, represents a complex network of GRK5-interacting factors that influence its protective and deleterious activities. Over the last decade, our group and others have focused attention on studying these molecules that influence the aforementioned activities of GRK5. We have tested specific approaches to block GRK5′s noxious effects in cardiomyocytes. These approaches go from pharmacological treatments to gene therapy approaches. Hence, in this review article, we will review the recent description of GRK5 involvement in cardiovascular and neurodegenerative disease and cancer. Moreover, we will inform the readers about the current state-of-the-art of targeting GRK5 to treat these disorders.

## 2. GRK5 Structure

All GRKs share a common and very conserved structure consisting of about 500 residues, with a short N-terminus (NT) region, that is unique to this family of kinases and implicated in GPCR binding, followed next by a regulator of G protein signaling (RGS) homology (RH) domain, and then the mostly conserved catalytic domain [6,21]. The latter is a Ser/Thr kinase domain (KD) that belongs to the AGC kinase family (e.g., PKA, PKC, PKG,) [18,22]. The final approximate one-third of the GRKs contain the CT region, which is involved in their membrane localization and which is quite different between GRK subfamilies [6,18,22]. GRK1 and 7 have short CT prenylation sequences [23], GRK2 and GRK3 contain pleckstrin homology (PH) domains that bind βγ subunits of G protein (Gβγ) and anionic phospholipids [17,24,25,26], while GRK4 and GRK6 carry CT palmitoylation sites and lipid-binding elements [10,27,28]. Although GRK5 shares a common structure with the other GRKs, it possesses several unique structural features. Indeed, GRK5 does not contain a palmitoylation site and rather it presents a group of hydrophobic amino acids within the CT domain of GRK5 that forms an amphipathic helix, that, along with phosphatidylinositol 4,5-bisphosphate (PIP2) binding, determines its membrane localization [6,29]. Importantly, as reported by Kunapuli et al., lipids contribute to GRK5 autophosphorylation (at the sites Ser 484 and Thr 485) and self-activation [30]. Recently, it was also reported that GRK5 can associate with the plasma membrane via NT residues 25–31 [31,32]. Furthermore, GRK5 localization to the membrane is regulated by the calcium sensor protein calmodulin (CaM). Importantly, CaM binding sites are located at each terminal domain, within residues 20–39 of GRK5′s NT and residues 540–578 of its CT [33], and binding of calcium to CaM and subsequently to GRK5 drastically impairs GRK5′s affinity for lipids and GPCRs [22,33,34]. Moreover, CaM binding promotes a different pattern of GRK5 autophosphorylation (Ser579, Ser583, and Ser584), which contributes to kinase inhibition [35]. PKC can also phosphorylate GRK5 on distinct sites in its CT (amino acids 565–572) that lead to inhibition of its catalytic activity [35].

GRK5 also contains specific caveolin binding motifs, and this characteristic is shared by GRK2 and GRK3, that through their PH (residues 567–584) and NT domains, interact with caveolins [36]. Besides, the GRK4 subfamily, including GRK5, contains a functional nuclear localization signal (NLS) (amino acids 388–395), which regulates the non-canonical functions of these kinases [16,37]. Interestingly, GRK5 also possesses a putative DNA-binding domain, which confers to this kinase a unique transcriptional regulator activity, over GRK4 and non-GRK4 subfamily [16]. Finally, a functional nuclear export sequence (NES, amino acids 259–265) has been identified within GRK5′s catalytic domain [16] (Figure 1). Since GRK5 is capable of interacting with proteins at the membrane, cytosol, or nuclear level, this kinase can influence different signaling pathways and can, therefore, be involved in several pathologic conditions, including cardiovascular diseases, neurodegenerative diseases, and cancer [38,39,40].

## 3. Cardiac Roles of GRK5

As mentioned above, GRK5 is anchored to the plasma membrane, exerting its canonical activities. These activities involve the phosphorylation and desensitization of activated GPCRs, including β-adrenergic receptors (βARs) that are well-recognized regulators of cardiac and neuronal function [41]. Importantly, several lines of evidence have suggested a role for GRK5 in the pathogenesis of cardiovascular disorders. For instance, Rockman et al. suggested that GRK5 represents one of the major culprits for βAR uncoupling in heart failure (HF). Indeed, in cardiac-specific overexpression of GRK5 (TgGRK5), these authors showed that GRK5 was responsible for marked desensitization of cardiac βARs in response to isoproterenol [42]. Subsequently, a role for GRK5 in HF development in humans has been reported. Indeed, high levels of GRK5 have been detected in human failing myocardium [43,44], in the left ventricles of patients with dilated cardiomyopathy [45], and in lymphocytes of patients with volume overload disease [46]. Importantly, these results were also confirmed in pre-clinical models of HF, as demonstrated by our research group, where TgGRK5 mice display enhanced cardiac hypertrophy and early HF compared with control mice in response to pressure overload, surgically induced by transverse aortic constriction (TAC) [47]. Conversely, cardiac-specific GRK5 knock-out (GRK5-KO) mice showed reduced hypertrophy and remodeling, along with preserved cardiac function following TAC or chronic administration of phenylephrine (α-AR agonist) [48]. Based on this premise, it is widely believed that the plasma membrane activity of GRK5 is deleterious due to receptor desensitization. However, several GRK5-dependent salutary effects have also been reported. For instance, Liggett and colleagues, by genetic analysis, observed that, in humans, among four single nucleotide polymorphisms (SNPs) identified in the *grk5* gene, only one, common in African Americans, in which leucine is substituted for glutamine at position 41, was able to protect mice against experimental catecholamine-induced cardiomyopathy [49]. Importantly, this GRK5 variant was found to prevent βAR hyperactivation, thus improving the survival rate of HF patients [49]. Other studies demonstrated a cardioprotective role for GRK5 acting as a bridge between two independent receptors, the βAR and the epidermal growth factor receptor (EGFR) [50]. In particular, Noma and coworkers demonstrated that in myocytes following β1AR stimulation, GRK5 can recruit β-arrestin and Src, leading to metalloproteinase (MMP) activation that in turn releases the heparin-bound EGF (HB-EGF) and transactivation of EGF receptor (EGFR). Following this, the EGFR is responsible for the activation of the mitogen-activated protein kinase ERK protective pathway, enhancing cell survival against chronic isoproterenol-induced cardiac damage [50]. For this reason, further investigation to better address the role of GRK5 at the plasma membrane is needed. As discussed above, since GRK5 has several distinct protein domains, it is crucial to better understand the role of each domain in determining signal transduction. In this regard, several studies have demonstrated that GRK5, thanks to its NLS, interacts with cytosolic or nuclear non-GPCR-related molecules [51,52]. Importantly, as demonstrated by us and others, through its nuclear localization, GRK5 can promote maladaptive cardiac hypertrophy and HF [16,47,53]. In accordance, mice overexpressing a mutant GRK5, without its functional NLS (Tg-ΔNLS), displayed less severe pathology in response to pressure overload compared to TgGRK5 mice [47]. Of note, in cardiac cells, nuclear accumulation of GRK5 occurs in response to stimulation of Gq-coupled receptors, particularly with phenylephrine or angiotensin II (AngII), and it is dependent on calcium/CaM binding to a specific site in GRK5 NT [54]. Indeed, a mutation within this site, negating CaM binding, was able to attenuate GRK5 nuclear translocation and hypertrophic gene transcription both in vitro, in cardiomyocytes, and in vivo, in mice [54]. Importantly, our studies have clarified that within the nuclei, GRK5 in response to pro-hypertrophic stimuli (e.g., phenylephrine, AngII, and pressure overload) can act as a class II histone deacetylase (HDAC) kinase, phosphorylating HDAC5 and leading to de-repression of MEF2-mediated hypertrophic gene transcription [47,53]. Moreover, in line with the report by Johnson and colleagues, we have also found that GRK5, via its unique DNA-binding activity, is able to induce pathological hypertrophic gene transcription [16]. GRK5 within the nucleus may bind to and regulate DNA independent of its kinase activity and act as a co-facilitator of the pathological transcription factor Nuclear factor of activated T-cells (NFAT) during cardiac hypertrophy [55]. In addition to the above, these non-canonical activities of GRK5 have recently demonstrated that this kinase can act downstream of aldosterone or its receptor, the mineralocorticoid receptor (MR). Particularly, in H9c2 cells (cardiomyoblasts), upon β2-AR activation, GRK5 can phosphorylate and inhibit the MR [56]. Conversely, in response to aldosterone, GRK5 translocates into the nucleus of the cardiomyocyte, promoting maladaptive pro-hypertrophic signaling [57]. Further studies have also added to the crucial role of GRK5 in promoting pathological cardiac hypertrophy. For instance, a recent study by Oda and colleagues demonstrated that, in cardiomyocytes, CaM released from the ryanodine receptor can migrate into the nucleus with GRK5 and promote HDAC5 export in response to AngII and phenylephrine, thus driving pro-hypertrophic signaling [58]. Recently, GRK5 activity has been found regulated by PH domain leucine-rich repeat protein phosphatase (PHLPP), a Ser/Thr phosphatase involved in the dephosphorylation of several kinases. Interestingly, PHLPP was able to interact with GRK5 to modulate phenylephrine-induced nuclear accumulation of GRK5 and cardiac hypertrophy in neonatal rat ventricular myocytes (NRVMs) [59].

## 4. GRK5 and Neurodegenerative Diseases

GRK5 has also been detected in the brain, in the limbic system [15,60,61], thus supporting a role for this kinase in controlling a variety of neuronal functions (e.g., memory, cognitive, and behavior). Moreover, as demonstrated by Chen and coworkers, GRK5 is involved in neuronal morphogenesis and in the establishment of functional neuronal circuitry [62]. Importantly, part of these effects is related to the ability of GRK5 to interact and regulate muscarinic cholinergic receptors, which are important GPCRs involved in the regulation of normal cognitive function [63,64]. In fact, Gainetdinov and colleagues demonstrated that GRK5 depletion in mice resulted in elevated sensitivity to cholinergic stimulation and impaired muscarinic receptor desensitization, with subsequent behavioral alterations [65]. In another report, Niu et al. showed that GRK5, in mice, controls social recognition via mTORC1 signaling [66]. As reported by Singh and coworkers, global GRK5-KO mice presenting with intermittent hypoxia display social behavior impairments, including cognitive decline [67]. One of the first demonstrations of GRK5-dependent effects on cognitive impairment was reported in 2004 by Suo et al. [68] These authors observed an abnormal GRK5, and GRK2, subcellular distribution closely associated with very early accumulation of soluble β-amyloid (Aβ) in the brain of Alzheimer’s disease (AD) transgenic models. Later, in 2007, Suo and colleagues also demonstrated that aged global GRK5-KO mice displayed selective working memory impairment, supporting the crucial role of GRK5 deficiency in the pathogenesis of Alzheimer’s disease (AD) [68,69]. AD is a neurodegenerative disorder, characterized by neuronal loss and cognitive decline [70]. At a molecular level, Aβ plaques, neurofibrillary tangles (which consist of insoluble aggregates of the hyperphosphorylated protein tau), and neuroinflammation are characteristic hallmarks of AD [71,72]. In addition to these features, several reports have demonstrated that the degeneration of cholinergic neurons represents an important pathogenic indicator of AD. To this extent, two pathogenic hypotheses, widely accepted and interconnected, have been proposed: the cholinergic and the amyloid cascade [73,74]. Based on this premise, since GRK5 activity/expression is correlated with both cholinergic dysfunction [65] and Aβ accumulation [69], it is more than plausible to consider this kinase as a key downstream player in these AD-related pathways. In line with this notion, Li et al. generated a novel mouse model of AD obtained by crossbreeding global GRK5-KO mice with a transgenic mouse line overexpressing a mutated version of Aβ precursor protein (Tg2576 or APPsw) [75]. In these mice, following an inflammatory trigger, such as the excessive deposition of fibrillar Aβ, GRK5 deficiency resulted in an exaggerated microgliosis and astrogliosis [75]. Interestingly, these effects appear to be gender-dependent, with female GRK5-KO mice displaying increased axonal defects and synaptic degeneration, compared to their male counterparts within this AD background [76]. Also, Liu et al. demonstrated that GRK5 deficiency was associated with the reduced hippocampal release of acetylcholine and desensitization of presynaptic M2/M4 muscarinic receptors with consequent cholinergic degeneration and cognitive decline [77]. The involvement of GRK5 in cholinergic dysfunction was also confirmed by Chen et al., who observed an increased Aβ accumulation in the brain and interstitial fluids in Tg2576/APPsw mice with GRK5 deficiency that was dependent on impaired cholinergic activity [78]. In line with these data, He at al. showed that GRK5 deficiency was also responsible for selective basal forebrain cholinergic neurodegeneration [79].

Importantly, GRK5 activity is mainly dependent on the subcellular compartment where it is located. In this context, Zhang and colleagues established that dysfunction in GRK5 subcellular localization might be a generic alteration associated with early AD. In details, these authors observed that GRK5 levels in the membrane fraction isolated from the hippocampus of aged APP(swe) mice were significantly decreased compared to the younger counterpart. At molecular levels, this membrane-associated GRK5 deficiency was associated with reduced desensitization of presynaptic muscarinic 2 receptors (M2) and with accelerated phosphorylation of tau [80]. Subsequently, Zhao et al. confirmed these results and also demonstrated that in the hippocampus of global GRK5-KO mice, the activity of both tau and GSK3β resulted increased and correlated with AD phenotype [81]. Finally, Zhang et al. demonstrated that in aged amyloid precursor protein (*APP*) and presenilin 1 (*PSEN1*) double transgenic mice (APP^+/−^/PS1^+/−^), the reduced levels of GRK5 on plasma membrane correlated with the onset of AD [82].

The importance of GRK5 in the pathogenesis of AD has also been suggested by studies in humans. In particular, genetic sequencing has identified two functional SNPs in the *grk5* gene, rs2230345 (GRK5-Q41L) and rs2230349 (GRK5-arginine [R] 304 histidine [H]) that have been linked to AD risk [82]. Specifically, Zhang et al. demonstrated in mice that the functional variant Q41L impaired the ability of GRK5 to translocate from the membrane to the cytoplasm and was associated with decreased levels of phosphorylated tau. Conversely, these authors demonstrated that the R304H variant enhanced the ability of GRK5 to translocate to the cytoplasm and to promote tau hyperphosphorylation. Additionally, in case-control studies in humans, they revealed that the Q41L variant is associated with a lower risk of late-onset AD [82]. Although the data described above suggest that GRK5 depletion contributes to neurodegeneration, Arawaka and colleagues have reported that high levels of GRK5 (both mRNA and protein) are accumulated in the brains of patients with Parkinson’s disease (PD) [83,84]. PD is the second most common neurodegenerative disease, after AD, and the most prevalent movement disorder [85]. At a molecular level, PD is characterized by the degeneration of dopaminergic neurons in the substantia nigra and by the formation of aggregates of α-synuclein in Lewy bodies [86]. Notably, GRK5 can phosphorylate α-synuclein at Serine in position 129, thus promoting its aggregation with the formation of soluble oligomers [83]. In 2010, Liu et al. demonstrated that α-synuclein enhanced the accumulation of GRK5 at the nuclear level, in both brain lysates from the transgenic mice model of α-synuclein overexpression, and in SH-SY5Y cells. Importantly, these authors showed that in the nuclei, GRK5 inhibited BCL-2 transcription and expression, thus increasing the apoptosis of neuronal cells [87]. Arawaka et al. demonstrated that in a Japanese population consisting of 286 patients with PD and 496 control subjects, the two functional SNPs (rs2420616 and rs4752293), in the introns of the *grk5* gene, were associated with sporadic PD [83]. Notably, these authors demonstrated that the DNA-binding transcription factor Yin Yang-1 (YY1) and the Cyclic adenosine monophosphate (cAMP) response element-binding protein 1 (CREB-1) bind to these functional SNPs and increased the transcriptional activity of the *grk5* gene [83]. Another study by Nicoletti et al. showed that the two polymorphic variants (rs2420616 and rs4752293) in the *grk5* gene and the related haplotypes play a critical role in predisposing patients with PD to cognitive impairment [88]. Despite this evidence, the role of GRK5 in PD remains controversial. For instance, a study from Tarantino and coworkers conducted in a cohort of PD patients of southern Italy (446 subjects with PD and 450 controls), investigating four SNPs (rs871196, rs2420616, rs7069375, and rs4752293) in the *grk5* gene, failed to find a correlation with the increased susceptibility to sporadic PD [89].

## 5. GRK5 and Cancer

Emerging evidence demonstrates that GPCRs and GRKs not only may play a pivotal role in cancer development and progression but also in its inhibition, depending on the cancer type and also on cellular localization [39,90,91]. As previously mentioned, GRKs are not only involved in modulating GPCR activities, but they can also interact with many non-GPCR proteins, including cell cycle regulators. In this regard, it has been shown that GRK5 is localized to the centrosome and is involved in the regulation of normal cell cycle progression [92]. Interestingly, at a nuclear level, GRK5 has been shown to phosphorylate nucleophosmin (NPM1), a nuclear protein that regulates cell cycle, centrosomal duplication, and apoptosis [93]. Importantly, GRK5-mediated phosphorylation of NPM1, at the level of serine in position 4, regulates sensitivity to polo-like Kinase 1 (PLK1) inhibitor-induced apoptosis, in both HeLa cells and breast cancer cells [93]. Another important link between GRK5 and tumors has been reported by studies showing the interaction between GRK5 and p53. Activation of p53 can occur in response to a variety of cellular stresses (e.g., DNA damage, hypoxia, and nucleotide deficiency) and leads to the transcriptional activation of several genes whose products are involved in cell-cycle arrest, DNA repair, or apoptosis. Importantly, p53 is highly regulated, through mechanisms of phosphorylation, dephosphorylation, and acetylation. Of note, alteration in such mechanisms can compromise cellular apoptotic response, thus contributing to abnormal cell growth and proliferation (tumorigenesis) [94,95]. Recent studies have reported that p53 activity/levels can be directly regulated by GRK5. For instance, Chen and coworkers demonstrated that both in vitro, in cultured osteosarcoma cells, and in vivo, in mice, GRK5 phosphorylates p53 at a threonine residue in position 55 (Thr-55) [96]. Of note, this phosphorylation promoted p53 ubiquitination with consequent inhibition of p53-dependent apoptotic response to DNA damage [96]. Analogously, the effects of GRK5 on p53 were confirmed a few years later by Okuda and Nishimura using nuclear magnetic resonance (NMR) spectroscopy monitored in real-time, showing GRK5-mediated phosphorylation of Thr-55 in the p53-acidic transactivation domain (TAD) [97]. In line with these data, different studies reported that GRK5 is also involved in the regulation of prostate tumor growth [98,99]. Specifically, Kim et al. demonstrated that prostate carcinoma cell line (PC3) presented higher levels of GRK5, compared to other cancer epithelial cell lines tested. Further, it was demonstrated that GRK5 silencing, via RNA interference (RNAi), impaired in vitro cell proliferation of PC3 cells. Analogously, the overexpression of a mutant form of GRK5 in which the kinase domain was inhibited, by means of substituting a Lysine with an Arginine in position 215 (kinase-dead GRK5-K215R mutant), resulted in PC3 cell-cycle arrest [98]. Analogously, Chakraborty and colleagues reported an important role for GRK5 in the regulation of prostate cancer cell migration and invasion. Importantly, they found that GRK5 was able to form a complex with and phosphorylate moesin, a protein that links membrane components to the actin cytoskeleton, thus regulating its cellular distribution. Notably, these authors reported that GRK5 silencing, in vivo in a xenograft model of human prostate cancer, resulted in a significant reduction of tumor growth, invasion, and metastasis [99]. There is also evidence for a significant role for GRK5 in brain cancers, as Kaur et al. demonstrated that this kinase was highly expressed in specimens of patients with grade II to grade IV glioma, and its expression was highly correlated with the aggressiveness of the tumor [100]. Furthermore, the expression and activity of GRK5 have been found altered in many other types of cancer cells. For example, it has been shown that GRK5 expression levels were significantly increased in patients with non-small-cell lung cancer (NSCLC) and correlated with worse survival rate, compared to patients with lower expression. Conversely, depletion of GRK5 inhibited in vitro NSCLC cell proliferation and in vivo formation of xenograft tumors [101]. In cervical and breast cancer cells, Lagman and coworkers demonstrated that GRK5 can phosphorylate HDAC6, increasing the resistance to the antitumor drug, paclitaxel [102]. In patients with renal cell carcinoma, increased levels of GRK5 were associated with poor clinical outcomes [102]. Importantly, as suggested by Zhao and colleagues, this effect was related to the enhanced proliferation of renal cell carcinoma cell lines in the presence of high GRK5 levels [103]. Finally, in rhabdomyosarcoma tumor cells, GRK5 has been proposed as a novel regulator of growth and self-renewal. Indeed, this kinase was able to regulate NFAT1 expression, in a kinase-independent manner, thus promoting tumor cell growth [104].

Despite these data, the role of GRK5 in tumor suppression appears ambiguous. Many studies have demonstrated that through the desensitization of GPCRs and non-GPCR-receptors, GRK5 can negatively affect cancer progression. Wu and coworkers have shown that in human colon cancer cells, Tazarotene-induced gene 1 (TIG-1)-mediated growth suppression is, at least in part, dependent on GRK5 [105]. Notably, in line with these results, Tsai et al. showed that GRK5 downstream TIG-1 inhibited prostaglandin E2 (PGE2) signaling, suppressing cancer cell proliferation [106]. In thyroid cancer cells, Métayé and colleagues demonstrated that GRK5 represents, together with GRK2, a major regulator of the thyroid stimulating hormone (TSH) receptor. GRK5-mediated TSH receptor desensitization resulted in a decreased response to TSH, with consequently reduced proliferation of thyroid cancer cells [107]. Finally, Geras-Raaka et al. showed that GRK5 induced the desensitization/downregulation of the Kaposi’s sarcoma (KS)-associated human herpesvirus (KSHV)-GPCR, inhibiting mouse fibroblasts proliferation [108]. KHSV is a gamma herpes virus with a genomic sequence containing several genes with likely oncogenesis-related functions. Many of these genes are viral homologs of cellular genes, but among these, there is one encoding for a GPCR (KSHV-GPCR) which has been demonstrated to induce cellular proliferation and transformation when overexpressed in murine cells [109] and to be involved in the pathogenesis of KS and primary effusion (or body cavity-based) lymphomas [108].

Therefore, GRK5 represents a novel regulator of cancer growth, exerting the opposite functions depending on subcellular localization. Indeed, when at the plasma membrane, GRK5 negatively affects tumor progression. In contrast, when in the nuclei and the cytosol, it enhances cancer progression. Its involvement in specific cancers needs to be better studied to determine the most appropriate targeting for potential therapeutic translation.

## 6. Targeting GRK5 as a Therapeutic Strategy for Chronic-Degenerative Disease

GPCRs mediate many physiological and pathological responses in different cell types [110,111]. Given the ability of GRKs to regulate not only GPCRs at the membrane but also intracellular proteins, targeting GRK5 could be an attractive therapeutic strategy in the treatment of cardiac and neuronal disease, as well as of cancer [39,112]. Since GRK2 and GRK5 are upregulated in the failing heart, their inhibition has been proposed as a novel and promising therapeutic approach [41,43,44]. Although there are several inhibitors for GRK2, far fewer selective compounds for GRK5 have been identified [113,114]. Recently, the development of selective covalent inhibitors, 5 and 16d, has been reported, that selectively target Cys474, a non-conserved cysteine typical of the GRK5 subfamily [115]. Since the ability of GRK5 to promote maladaptive cardiac hypertrophy and HF, several therapeutic strategies have been proposed to inhibit these harmful effects. In this regard, intracardiac delivery of an adenovirus encoding for the NT of GRK5 (GRK5-NT), which is implicated in GPCR binding and transcription regulation in the nucleus [22,116], has been recently proposed as a novel therapeutic approach to inhibit and reduce left ventricular maladaptive hypertrophy [117]. Indeed, as reported by Sorriento et al., cardiac overexpression of GRK5-NT inhibited NFAT-dependent GATA-4 activation, reducing cardiac hypertrophy both in vitro in H9c2 cardiomyoblasts and in vivo in spontaneously hypertensive rats [118]. Recently, it has been demonstrated that malbrancheamide, a natural product capable to bind the C-terminal domain of CaM, attenuated GRK5 translocation into the nucleus, thus blocking cardiac hypertrophy [119]. Moreover, KR-39038, a novel GRK5 small inhibitor, significantly inhibited cellular hypertrophy and HDAC5 phosphorylation in vitro in neonatal rat cardiomyocytes. Interestingly, this inhibitor was also able to reduce cardiac hypertrophy and preserve cardiac function in either mouse or rat models following TAC [120]. Another important compound recently proposed as a GRK5-selective inhibitor is an anti-inflammatory and anti-allergic immunomodulator approved by the Food and Drug Administration (FDA), called amlexanox. This compound has been reported as a low micromolar inhibitor of GRK5 [121]. In detail, the treatment of neonatal rat ventricular myocytes (NRVMs) with amlexanox (50 μM) was able to inhibit GRK5 consistently with MEF2 transcriptional activity [121]. In addition to the study of GRK5 inhibitors on cardiomyocytes, studies have explored the therapeutic potential of GRK5 inhibitors in cancer, and specifically, for the prevention of GRK5-dependent tumor progression. For example, Pham et al. have demonstrated that treatment of mice with human rhabdomyosarcoma xenografts, with CCG-215022, a selective GRK5 inhibitor, resulted in a significant reduction of tumor growth [104]. Moreover, a recent study by Sommer et al. investigated the effect of sunitinib, a potent small-molecule inhibitor, on GRK5 and breast cancer cells [122]. They demonstrated that in vitro treatment with sunitinib decreased GRK5 expression and migration in a cancer cell line [122]. GRK5 inhibitors are summarized in Table 1.

## 7. Conclusive Remarks

In the last decades, GRK5 activities have been extensively studied in heart disease. However, since it is generally considered to be ubiquitously expressed in mammalian tissues and its activities are multiple and cell type- and receptor-specific, as demonstrated in the brain and cancer, it is very important to determine whether and how the extra-cardiac activities of GRK5 can influence all mammalian physiology.

Previous in vitro and in vivo data in rodents supported an important role played by GRK5 in the pathophysiology of atherosclerosis, though, either pro-atherogenic or anti-atherogenic activities [123]. Further, GRK5 has been shown to influence metabolism in mice since its deficiency resulted in decreased weight gain and white adipose tissue (WAT) mass, reduced transcription of adipogenic genes, and inhibited adipocyte differentiation [124,125]. In line with these data, a genome-wide association study (GWAS) in a Chinese population found a significant relationship between an intronic SNP rs10886471 of grk5 gene and type 2 diabetes [126]. Subsequently, Lutz et al. observed that one SNP in intron 3 (rs10886471) can affect cardiometabolic traits, in a cohort of 2332 unrelated Caucasians at risk for type 2 diabetes, from the southern part of Germany [127]. Of note, these authors concluded that carriers of the minor allele of rs10466210 displayed a trend for higher intima-media thickness of the carotid artery, supporting the targeting of GRK5 as a novel strategy aiming at modifying cardiovascular risk.

In this review article, we have also detailed the impact of GRK5 on both AD and PD. However, we cannot exclude the potential implications of GRK5 in other neuronal disorders. For example, a report by Franklin et al. supported a role for GRK5 in the onset of neuropsychiatric disorders. Indeed, these authors demonstrated that GRK5 is involved in the cannabinoid-induced upregulation and enhanced activity of serotonin 2A (5-HT_2A_) receptors in neuronal cells, and dysfunction of these receptors may enhance the onset of mental disorders such as anxiety, psychosis, and schizophrenia [128]. Further, Wang et al., in a total of 344 Taiwanese patients under methadone maintenance treatment, demonstrated a role for GRK5 in the regulatory mechanisms of methadone dose and course of heroin dependence [129]. Finally, as discussed above, the role of GRK5 in tumor growth appears relevant but also somewhat controversial. Indeed, when GRK5 is localized at the plasma membrane, it mostly exerts an anti-tumor effect. However, when GRK5 is in the cytosol or nucleus, it can promote tumor growth. Thus, the comprehension of how the subcellular localization drives the effects of GRK5 is utterly necessary. In this regard, as also discussed in a study by Zhao et al. [81], the lack of molecules able to target GRK5 in specific subcellular compartments limits our understanding of whether and how a deficit and plenty in GRK5 can be considered a generic pathologic alteration. It is equally important to define how the protein domains of GRK5 and the associated interactome activate the canonical, rather than non-canonical, pathways. For instance, more efforts should be made to understand the specific effects of the interaction between GRK5 and p53. Indeed, p53 is involved in several biological and pathological processes and recently, it has been associated with the regulation of ferroptosis [130]. This iron-dependent new form of programmed cell death is driven by lipid peroxidation and is involved in cancer, tissue ischemia/reperfusion injury, and neurodegeneration [130,131]. Whether and how GRK5 via p53 regulates ferroptosis is an urgent question that deserves an immediate response. Thus, these studies will help us in generating a strategy to specifically target this kinase and block all undesired harmful effects, rather than the protective ones.

In conclusion, in the present study, we have reviewed the current state-of-the-art of GRK5 biology and its involvement in cardiovascular, neurodegenerative, and neoplastic disorders (Figure 2). Despite the evidence provided above, the exact mechanism involved in many of the untoward effects of GRK5 remains uncertain. However, all these translational questions will be answered, with the advent of novel molecules (like those displayed in Table 1) that can selectively inhibit GRK5-dependent toxic effects.

## Figures and Tables

**Figure 1 ijms-22-01920-f001:**
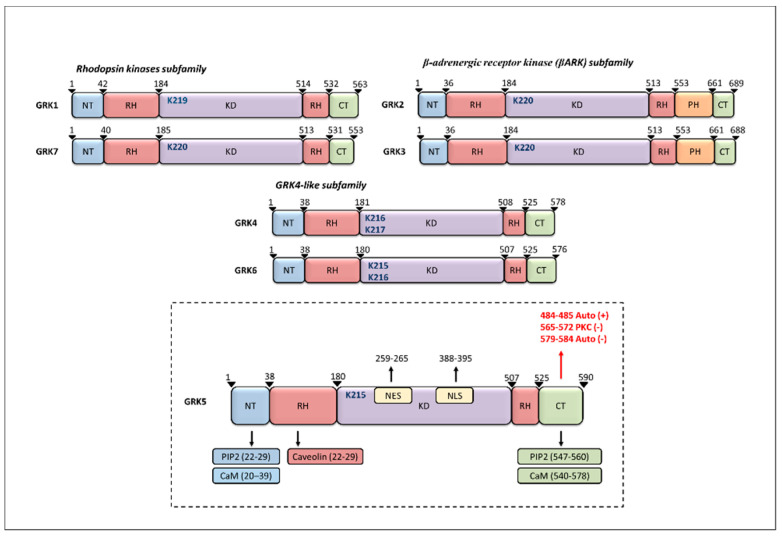
Schematic representation of G protein-coupled receptor kinases (GRKs) structure. All GRKs share a common structure, comprising a N terminal (NT) region, followed by an RH domain, which includes the kinase domain (KD). Catalytic activity is dependent on lysine residues (in blue) that are specific for each isoform. The C terminal (CT) region is variable among the different subfamilies. GRK5 structure includes indication of phosphatidylinositol 4,5-bisphosphate (PIP2), Calmodulin (CaM), and caveolin binding sites. The residues of GRK5 that are autophosphorylated or phosphorylated by protein kinase C (PKC) are indicated in red.

**Figure 2 ijms-22-01920-f002:**
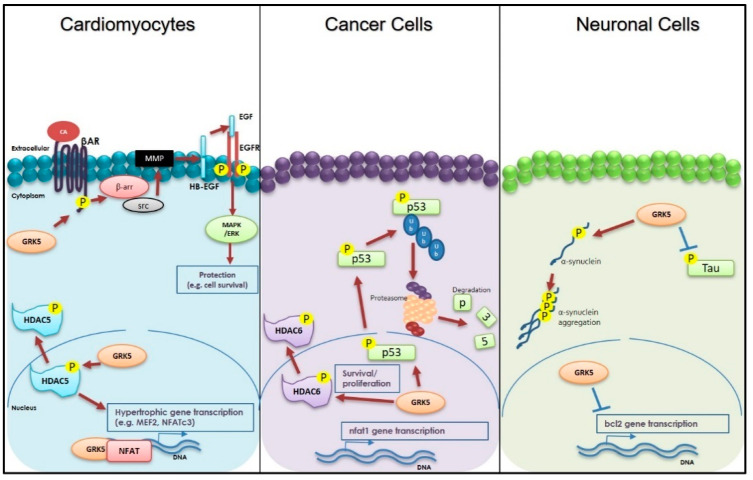
Schematic representation of G protein-coupled receptor (GPCR) kinase 5 (GRK5) activities within cardiomyocytes, cancer, and neuronal cells. In cardiomyocytes: GRK5 is involved in phosphorylation of GPCRs, including the β-adrenergic receptor (βAR). Importantly, following catecholamine (CA) stimulation of βAR, GRK5 is also able to induce the transactivation of the epidermal growth factor (EGF) receptor (EGFR) via β-arrestin (β-arr)/src/matrix metalloproteinases (MMP) pathway activation. EGFR in turn activates a Mitogen-activated protein kinase (MAPK)/ extracellular-signal-regulated kinase (ERK) protective pathway. In the nuclei, GRK5 is able to either bind to and phosphorylate Histone deacetylase 5 (HDAC5) or bind the DNA and Nuclear factor of activated T-cells (NFAT), thus enhancing hypertrophic gene transcription. In cancer cells: GRK5 regulates p53 via direct phosphorylation and degradation. Moreover, into the nuclei, GRK5 can phosphorylate HDAC6 or can increase NFAT1 expression. In neuronal cells: GRK5 inhibits Tau phosphorylation. Conversely, this kinase can increase α-synuclein phosphorylation, promoting its aggregation. Finally, in the nucleus, GRK5 inhibits B-cell lymphoma 2 (bcl2) gene transcription, increasing the apoptosis.

**Table 1 ijms-22-01920-t001:** Summarizing known GRK5 inhibitors, including the type of pre-clinical studies and their structures. The chemical structures were produced on ChemDraw (accessed on 14 February 2021).

GRK5 Inhibitors	Type of Study	Structure
Compound 5	In vitro (Rowlands et al., 2019)	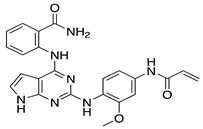
Compound 16d	In vitro (Rowlands et al., 2019)	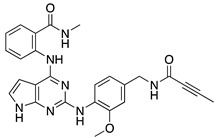
Malbrancheamide	In vitro (Beyett et al., 2019)	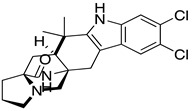
KR-39038	In vitro/In vivo (Lee et al., 2020)	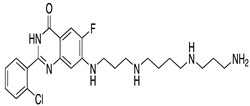
Amlexanox	In vitro (Homan et al., 2014)	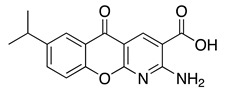
CCG-215022	In vitro/In vivo (Pham et al., 2020)	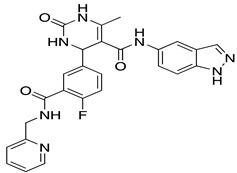
Sunitinib	In vitro (Sommer et al., 2019)	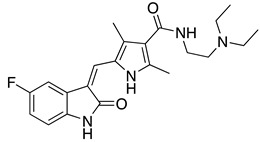
Adenovirus (Ad) GRK5-NT	In vitro/In vivo (Sorriento et al., 2010)	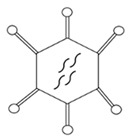

## Data Availability

Not applicable.

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
