# Peer review of "Targeting GRK5 for Treating Chronic Degenerative Diseases"

_ijms, 2021, doi:10.3390/ijms22041920_

Round 1

Reviewer 1 Report

I have reviewed the manuscript entitled ''Grk5 Biology and Targeting in Chronic-Degenerative Disorders'' - the manuscript deserves a major revision before getting it accepted

  1. The manuscript deserves substantial editing and proofreading. For example, several typographical errors have been found, line 20: ''role'' instead of ''rule''. The title needs to be modified - the suggested title: Targeting Grk5 biology for treating chronic degenerative diseases. Several unnecessary phrases and connecting words have been used, for example, line 29: ''For this reason'' line 238: importantly, line 239: Molecularly
  2. In the introduction general definition of GRKs should be given first after GPCRs. 
  3. Line 49-50 are unclear - ''ubiquitously expressed'' and ''more restricted expression pattern'' unclear
  4. Paragraph 2 and 3 in the introduction should be merged, and the last paragraph in the induction could be improved. 
  5. A general figure on GRK structure should be added. 
  6. ''GRK5 and Neurodegenerative diseases'' need to be work more on the association between the information - a better story needs to be formed.
  7. ''GRK5-R304H and Q41L mutations'' are they studied with any APP and PS mutations?  
  8. Is there any literature on GRKs role in other neurodegenerative diseases? If no, authors should be explained about challenges and prospects
  9. ''GRK5 as an oncogenic factor'' and ''Targeting GRK5 as a Therapeutic Strategy for chronic-degenerative disease'' - the story could be better. A descriptive table should be given with the chemical structure of inhibitors and activators.  
  10. The conclusion also needs to be imprroved.

Author Response

We thank the Reviewer for His/Her constructive feedback and comments on our manuscript. We have conducted an extensive revision to address the Reviewer's major and minor points. We trust that the responses and corresponding changes made to the revised manuscript, as detailed below, have satisfactorily addressed the issues raised; and that our revised manuscript will be now deemed suitable for publication.

Major concerns

  1. The manuscript deserves substantial editing and proofreading. For example, several typographical errors have been found, line 20: ''role'' instead of ''rule''.

Reply. We apologize for this typographical error. We corrected as indicated. Thanks

  1. The title needs to be modified - the suggested title: Targeting Grk5 biology for treating chronic degenerative diseases.

Reply: Thanks for this suggestion. Accordingly, we changed the title.

  1. Several unnecessary phrases and connecting words have been used, for example, line 29: ''For this reason'' line 238: importantly, line 239: Molecularly

Reply: Thanks for your suggestion. We removed the unnecessary words.

  1. In the introduction general definition of GRKs should be given first after GPCRs.

Reply: We are now describing GRKs before GPCRs. Thanks.

  1. Line 49-50 are unclear - ''ubiquitously expressed'' and ''more restricted expression pattern'' unclear

Reply: We thank the Reviewer for raising this concern. Accordingly, We have revised the manuscript. Please see page 1 line 44.

  1. Paragraph 2 and 3 in the introduction should be merged, and the last paragraph in the induction could be improved.

Reply: We thank the Reviewer for their comments.  Paragraphs 2 and 3 have been merged and the last paragraph of the introduction has been improved.

  1. A general figure on GRK structure should be added.

Reply: This is another good point. We have included a new figure (fig. 1) in the revised version of the manuscript. Thanks.

  1. ''GRK5 and Neurodegenerative diseases'' need to be work more on the association between the information - a better story needs to be formed.

Reply: We completely revised and improved the paragraph as suggested. Thanks.

  1. ''GRK5-R304H and Q41L mutations'' are they studied with any APP and PS mutations?

Reply: We really thank the Reviewer for his/her comments. We have included this information in the new version of the manuscript. Please see page 6 line 258.

  1. Is there any literature on GRKs role in other neurodegenerative diseases? If no, authors should be explained about challenges and prospects

Reply: We really thank the Reviewer for raising this comment. In the new version of the manuscript we have included a new paragraph entitled: 7. Future perspectives. Please see page 11

  1. ''GRK5 as an oncogenic factor'' and ''Targeting GRK5 as a Therapeutic Strategy for chronic-degenerative disease'' - the story could be better. A descriptive table should be given with the chemical structure of inhibitors and activators.

Reply: Thanks for this comment. To better address the Reviewer’s concerns, we have improved both paragraphs and we added a table as indicated.

  1. The conclusion also needs to be improved.

Reply: We thank the Reviewer for their comments. We have now included a new ending paragraph entitled Future perspectives and have also improved the conclusions.

Reviewer 2 Report

This review attempted to describe the role of GRK5 in cardiac hypertrophy and heart failure, neurodegenerative diseases, and cancer. The subject is important for the field, and a good review would be timely. Unfortunately, the authors provided no conceptual framework that would justify a review. In fact, they did not move beyond “these authors showed this, while these authors showed that”. The authors did not even mention certain GRK5-specific features, such as lipid-induced self-activation by phosphorylation described 26 years ago (J Biol Chem 1994 Apr 8;269(14):10209-12). They did not mention that other GRKs, e.g., GRK2 and GRK3, also phosphorylate many non-GPCR substrates, thereby affecting GPRC-independent processes in the cell. The authors cite correlations between intronic SNPs in GRK5 gene and diseases, but do not offer any idea how intronic changes could have affected GRK5 function. There are also numerous factual errors in the text (some indicated below, but these are too numerous to point them all out). Grammar and word usage are both sub-standard.

Extensive editing, preferably by a native speaker familiar with the GPCR/GRK field, is needed. Numerous corrections in grammar, word usage, and substance are necessary. Just a few examples out of many: line 37, delete “in details”; line 39, Gbg is a dimer, not a monomer; lines 57-58, all GRKs have three domains (as correctly acknowledged in line 72), the structure of GRK5 is typical for GRK4 subfamily; line 60, all GRKs are multi-functional, e.g., GRK2 interacts with GPCRs, Gbg, and Ga of Gq/11 subfamily, and numerous non-GPCR substrates, there is even structure of GRK2 in three-protein complex with Gbg and Gaq (Science. 2005 Dec 9;310(5754):1686-90); line 73, GRKs have nothing “extracellular”, they are cytoplasmic proteins, GRCRs, not GRKs, have an extracellular N-terminus; line 76, RGS homology domain in all GRKs consists of two parts, large part preceding the kinase domain, and smaller part following it (see numerous available structures, or ref 32 for review); lines 189-90: neurofibrillary tangles consist of hyperphosphorylated protein tau, so these are not two different features; etc.

Author Response

We thank the Reviewer for His/Her constructive feedback and comments on our manuscript. We have conducted an extensive revision to address the Reviewer's major and minor points. We trust that the responses and corresponding changes made to the revised manuscript, as detailed below, have satisfactorily addressed the issues raised; and that our revised manuscript will be now deemed suitable for publication.

Major issues

  1. the authors provided no conceptual framework that would justify a review. In fact, they did not move beyond “these authors showed this, while these authors showed that”.

REPLY: We thank the Reviewer for raising these concerns. We have completely revised the manuscript and provided a conceptual framework as suggested.

  1. The authors did not even mention certain GRK5-specific features, such as lipid-induced self-activation by phosphorylation described 26 years ago (J Biol Chem 1994 Apr 8;269(14):10209-12).

Reply: We thank the Reviewer for pointing out this missing information. As indicated, we have now included this information in the revised version of the manuscript. Please see page 3 line 101.

  1. They did not mention that other GRKs, e.g., GRK2 and GRK3, also phosphorylate many non-GPCR substrates, thereby affecting GPRC-independent processes in the cell.

REPLY: This is another good point. We are now including this information in the new version of the manuscript (please see page 2 line 65). Thanks

  1. The authors cite correlations between intronic SNPs in GRK5 gene and diseases, but do not offer any idea how intronic changes could have affected GRK5 function.

REPLY: We apologize with the Reviewer for this missing information. Thus, we have added this information. Please see page 7 line 285; page 11 line 447. Thanks

  1. There are also numerous factual errors in the text (some indicated below, but these are too numerous to point them all out). Grammar and word usage are both sub-standard.

REPLY. We really thank the Reviewer for his/her precious comments. We have completely revised the grammar and word usage as suggested. The manuscript was revised by Dr. Koch who is a native speaker and an expert in the GRK field.

  1. Extensive editing, preferably by a native speaker familiar with the GPCR/GRK field, is needed. Numerous corrections in grammar, word usage, and substance are necessary. Just a few examples out of many: line 37, delete “in details”

Reply: We have corrected the grammar as indicated.

  1. line 39, Gbg is a dimer, not a monomer;

Reply: We apologize for this mistake. We have completely revised the paragraph.

  1. lines 57-58, all GRKs have three domains (as correctly acknowledged in line 72), the structure of GRK5 is typical for GRK4 subfamily;

Reply: We really thank the Reviewer for raising these concerns. We have completely revised the paragraph “GRK5 structure”. Please see page 2 line 86.

  1. line 60, all GRKs are multi-functional, e.g., GRK2 interacts with GPCRs, Gbg, and Ga of Gq/11 subfamily, and numerous non-GPCR substrates, there is even structure of GRK2 in three-protein complex with Gbg and Gaq ( 2005 Dec 9;310(5754):1686-90);

Reply: We have included this info and references as indicated. Please see page 2 line 55. 96. Thanks.

  1. line 73, GRKs have nothing “extracellular”, they are cytoplasmic proteins, GRCRs, not GRKs, have an extracellular N-terminus;

Reply: We apologize for this mistake. We have corrected as indicated.

  1. line 76, RGS homology domain in all GRKs consists of two parts, large part preceding the kinase domain, and smaller part following it (see numerous available structures, or ref 32 for review);

Reply: Please see our reply above.

  1. lines 189-90: neurofibrillary tangles consist of hyperphosphorylated protein tau, so these are not two different features; etc.

Reply: We have corrected as follows: neurofibrillary tangles which consist of insoluble aggregates aggregates of hyperphosphorylated protein tau.

Reviewer 3 Report

This review article by Marzano et al. covers an interesting and important topic. The essential role of GRKs for shaping the signalling repertoire of GPCRs have been overlooked for a long time. There is a rising interest in GRKs, not only in GPCR-dependent signaling. The authors focus on GRK5 and its role in chronic-degenerative disorders. This focus might be too narrow for an isolated article, but since its part of a special issue on GRKs, this specialization becomes reasonable. However it would be beneficial for the reader to briefly touch some more general points:
For example, the second paragraph on the structural aspects could be a little bit more detailed.
Another suggestion would be to briefly mention the role of GRK5 in other diseases/therapeutic fields, e.g. opioid analgesics.
If, these aspects will be covered in a revised version, I would like to recommend publication of this nicely written review.

Author Response

We thank the Reviewer for His/Her constructive feedback and comments on our manuscript. We have conducted an extensive revision to address the Reviewer's major and minor points. We trust that the responses and corresponding changes made to the revised manuscript, as detailed below, have satisfactorily addressed the issues raised; and that our revised manuscript will be now deemed suitable for publication.

Major issues

  1. The second paragraph on the structural aspects could be a little bit more detailed.

Reply: We thank the Reviewer for raising this concern. We have completely revised the paragraph “GRK5 structure”.

  1. Another suggestion would be to briefly mention the role of GRK5 in other diseases/therapeutic fields, e.g. opioid analgesics.

Reply: We have discussed this role in the new paragraph “Future Perspectives”. Please see page 11.

Round 2

Reviewer 1 Report

Authors have been revised the manuscript very well. However, the manuscript still deserves some minor modifications.

This is good authors have added a table, and the provided chemical structure should be produced by software like ChemDraw. 

Future perspective and conclusion should be merged into one section. 

Ferroptosis, a novel form of, iron-mediated cell death implicated in the progression of various diseases and disorders such as cancer and neurological disorders. Can the authors narrate on this cell death pathway? How could Grk5 be implicated with the ferroptosis? 

Author Response

We are very thankful for the revision process and the useful points raised by this Reviewer which have allowed us to strengthen the quality of our manuscript. We have conducted another revision to address the Reviewer’s minor points. We trust that the responses and corresponding changes made to the revised manuscript, as detailed below, have satisfactorily addressed the issues raised; and that our revised manuscript will be now deemed suitable for publication in International Journal of Molecular Sciences.

 1. This is good authors have added a table, and the provided chemical structure should be produced by software like ChemDraw.

Reply: We thank the reviewer for this comment. The chemical structures of GRK5 inhibitors have been produced using a software available on ChemSpider.com

2. Future perspective and conclusion should be merged into one section.

Reply: Thanks for this suggestion.  Accordingly, we have merged both paragraphs into one section.

3. Ferroptosis, a novel form of, iron-mediated cell death implicated in the progression of various diseases and disorders such as cancer and neurological disorders. Can the authors narrate on this cell death pathway? How could Grk5 be implicated with the ferroptosis?

Reply:  We thank the Reviewer for this comment. We have discussed the possibility that GRK5 can be involved in such process via p53 regulation. Please see our discussion at page 13 line 488.

Reviewer 2 Report

The review was thoroughly re-written and became a lot more coherent. Now it is clear what authors’ messages are. A few editorial changes would further improve the manuscript.

  1. Title: delete the word “biology”. You can target GRK5, but not “biology”.
  2. Lines 353-360. If one takes this at face value, both GRK5 and its deficit increase hyperphosphorylation of tau and AD development. Which is true?
  3. Some editing is still needed: line 59, delete “respectively”, as GRK1 is also present in cones; line 211, delete “of research”, or write “our research group”; line 329, replace “further” with “in addition”; lines 348 and 351, replace “by” with “on”; line 5712, delete “via”; line 638, “all mammal’s physiology” should be “mammalian physiology”. Also, phrases like “as discussed above” in most cases can be deleted to make the text easier to read w/o affecting contents. Some grammatically awkward expressions should be rephrased.

Author Response

We are very thankful for the revision process and the useful points raised by the Reviewer which have allowed us to strengthen the quality of our manuscript. We have conducted another revision to address the Reviewer’s minor points. We trust that the responses and corresponding changes made to the revised manuscript, as detailed below, have satisfactorily addressed the issues raised; and that our revised manuscript will be now deemed suitable for publication in International Journal of Molecular Sciences.

Title: delete the word “biology”. You can target GRK5, but not “biology”.

Reply: As indicated we have modified the title. Thanks

Lines 353-360. If one takes this at face value, both GRK5 and its deficit increase hyperphosphorylation of tau and AD development. Which is true?

Reply: We apologize to the Reviewer for this misunderstanding. Thus, we have revised the paragraph (Please see page 6 line 252). In this new version, we have discussed that the findings by Zhang et al. and Zhao et al. were both related to a membrane-associated GRK5 deficiency. However, in the study by Zhao et al. (2018) it was also demonstrated that global deficiency of GRK5 can influence tau phosphorylation and AD development.  Thanks for your comments.

Some editing is still needed: line 59, delete “respectively”, as GRK1 is also present in cones; line 211, delete “of research”, or write “our research group”; line 329, replace “further” with “in addition”; lines 348 and 351, replace “by” with “on”; line 5712, delete “via”; line 638, “all mammal’s physiology” should be “mammalian physiology”. Also, phrases like “as discussed above” in most cases can be deleted to make the text easier to read w/o affecting contents. Some grammatically awkward expressions should be rephrased.

REPLY. We thank the Reviewer for these comments. We have revised the manuscript as indicated.